# Spatially Explicit Mapping of Soil Conservation Service in Monetary Units Due to Land Use/Cover Change for the Three Gorges Reservoir Area, China

**Shicheng Li** [1,2] **, Zilu Bing** [1] **and Gui Jin** [3,*]

1   Department of Land Resource Management, School of Public Administration, China University of Geosciences, Wuhan 430074, China; lisc@cug.edu.cn (S.L.); bingzilu@cug.edu.cn (Z.B.)
2   Shandong Provincial Key Laboratory of Depositional Mineralization and Sedimentary Mineral, College of Earth Science and Engineering, Shandong University of Science and Technology, Qingdao 266590, China
3   College of Urban and Environmental Sciences, Central China Normal University, Wuhan 430079, China
*   Correspondence: jingui@igsnrr.ac.cn

**Abstract:** Studies of land use/cover change (LUCC) and its impact on ecosystem service (ES) in monetary units can provide information that governments can use to identify where protection and restoration is economically most important. Translating ES in monetary units into decision making strongly depends on the availability of spatially explicit information on LUCC and ES. Yet such datasets are unavailable for the Three Gorges Reservoir Area (TGRA) despite its perceived soil conservation service value (SCSV). The availability of remote sensing-based datasets and advanced GIS techniques has enhanced the potential of spatially explicit ES mapping exercises. Here, we first explored LUCC in the TGRA for four time periods (1995–2000, 2000–2005, 2005–2010, and 2010–2015). Then, applying a value transfer method with an equivalent value factor spatialized using the normalized difference vegetation index (NDVI), we estimated the changes of monetary SCSV in response to LUCC in a spatially explicit way. Finally, the sensitivity of SCSV changes in response to LUCC was determined. Major findings: (i) Expansion of construction land and water bodies and contraction of cropland characterized the LUCC in all periods. Their driving factors include the relocation of residents, construction of the Three Gorges Dam, urbanization, and the Grain for Green Program; (ii) The SCSV for TGRA was generally stable for 1995–2015, declining slightly (<1%), suggesting a sustainable human–environment relationship in the TGRA. The SCSV prevails in regions with elevations (slopes) of 400–1600 m ($0°$–$10°$); for Chongqing and its surrounding regions it decreased significantly during 1995–2015; (iii) SCSV's sensitivity index was 1.04, 0.53, 0.92, and 1.25 in the four periods, respectively, which is generally low. Chongqing and its surrounding regions, with their pervasive urbanization and dense populations, had the highest sensitivity. For 1995–2015, 70.63% of the study area underwent increases in this sensitivity index. Our results provide crucial information for policymaking concerning ecological conservation and compensation.

**Keywords:** ecosystem service; economic valuation; equivalent value factor; different construction periods; human activities; Yangtze River Basin

## 1. Introduction

Ecosystems provide a range of services, many of which are of fundamental importance to human well-being and livelihoods [1]. However, human land use activities have changed land cover dramatically and led to significant changes in the services provided by ecosystems to humans [2–5]. As important components of human well-being, LUCC (land use/cover change) and its impact on ecosystem services (ES) has received extensive attention globally [1,6–8]. To investigate this,

several scientific programs have been launched, including the Millennium Ecosystem Assessment (http://www.millenniumassessment.org/), Global Land Programme (https://glp.earth/), Mapping and Assessment of Ecosystems and their Services (MAES) in Europe, The Economics of Ecosystems and Biodiversity (http://www.teebweb.org/), and the Intergovernmental Science-Policy Platform on Biodiversity and Ecosystem Services (https://www.ipbes.net/). Because of these international, collaborative science programs, the assessment and mapping of ES has made great strides [1].

Valuation of ES in monetary units based on land use/cover maps is a crucial approach central to ES mapping [7]. It is appropriate for geographical areas where data availability is limited, in contrast to mapping based on process-based ecosystem models [9,10], providing additional information to public entities for identifying where protection and restoration efforts are economically most important, on which they could base policy concerning ecological conservation and compensation schemes. The seminal work by Costanza et al. [11] on the total value of global ES was a milestone in the mainstreaming of ES [1,7]; since then, there has been a steady growth in the number of publications addressing the monetary valuation of natural resources, ES, and biodiversity [12–16], with those same authors updating their estimates for 1997 and 2011 [17]. Following this approach, many estimations of ES value in monetary units at the regional scale were conducted [1,18–21]. In China, based on a questionnaire survey of more than 700 scholars with expertise in ecology, Xie et al. [22] proposed a representative ES evaluation unit system and later updated its parameters [13,23]. In addition, de Groot et al. [7] developed an ES value database holding more than 1350 value-estimates in monetary units, which is open access and allows researchers to both retrieve and submit new data. Compared with spatially explicit mapping in biophysical terms, however, most of these monetary assessments are administrative division-based valuations [14,16,20]; hence, they cannot reflect fine-scale details of spatial patterning and variation in ES, limiting their widespread applicability [24].

As the world's largest power station in terms of installed capacity, the Three Gorges (the Qutang, Wu, and Xiling gorges) Project (TGP) has greatly impacted both the biodiversity and ES of the Three Gorges Reservoir Area (TGRA), alarming many ecologists worldwide [25,26]. As a mountainous region, the TGRA is one of the most ecologically valuable regions in China, since it provides substantial soil conservation service value (SCSV) [27,28] and has a prominent role in maintaining ecological health. Since the onset of this dam's construction, significant human land use activities and land degradation in this region has ensued, in the name of economic development [29]. In recent years, however, the Chinese government has come up with a series of sustainable policies to restore and conserve the ecological environment of the Yangtze River Basin [30,31], especially for the TGRA which features strong human versus nature conflicts. Therefore, it is important that we study LUCC and its impacts on ES, especially the SCSV [28], for the TGRA. The findings of this study will contribute to the theoretical basis of ecological protection and compensation and empirically inform the restoration and protection of the environment in the Yangtze River Basin. Not surprisingly then, many studies have sought to identify the characteristics of LUCC [32] and its influence on either ES or SCSV for the TGRA in recent decades [33,34].

The spatial-temporal patterns, changing characteristics, and main drivers of LUCC during 1955–2000 for the TGRA have already been analyzed [35]. For the 1990–2015 period, the TGP's construction and associated mass relocation of residents/towns/villages, economic development, and urbanization led to the continuous expansion of artificial surfaces and wetlands in the TGRA [35–37]. But the specific characteristics and driving factors of LUCC and its ecological effects for different construction periods of TGP are either missing or not yet fully explored; lacking this knowledge hinders our ability to devise the most applicable adaptation, protection, and sustainable development measures for future. Some studies suggest that LUCC in the TGRA has considerably affected biodiversity and ES through habitat fragmentation [25,26,38,39]. In terms of ES or SCSV, the study of Yan et al. [27] demonstrated that monetary ES value at the county scale for the TGRA during 1990–2011 gradually increased, whereas Guo et al. [40] showed that for 2000–2014 period it decreased because cropland/forestland was converted to construction land. Employing the Integrated Valuation of

Ecosystem services and Trade-offs (InVEST) model, Xiao et al. [33] assessed how SCSV was altered in the TGRA because of land use change for 2000–2010, finding that 83.4% of the total area experienced an increase in SCSV. For the same time period, Xiong et al. [41] evaluated the changes in SCSV's spatiotemporal patterns, by using the universal soil loss equation [42,43], which revealed great spatial heterogeneity of SCSV in the TGRA. Further related studies were particular to the Chongqing [28,44–46] or Hubei [47,48] sections of the TGRA. Generally, the outputs of most SCSV valuations for TGRA are made in biophysical terms (e.g., tons of soil retention/loss), which require extensive primary data whose collection is often time-consuming and expensive. Consequently, such studies only cover a short time period or parts of the TGRA. Although several valuations in monetary units [27,40] were conducted at county level, or more coarse levels, they cannot provide sufficient spatial information to help decision makers in their trade-off analyses between economic benefits and conservation of ES. Moreover, little ES-mapping research has investigated the differences among ecological effects from LUCC for successive construction periods of the TGP.

In this context, the aims of this study were threefold: (1) To explore the specific characteristics and driving factors of LUCC in the TGRA for four construction periods (1995–2000, 2000–2005, 2005–2010, and 2010–2015) of the TGP; (2) To spatially explicit map the dominate ES in the TGRA (i.e., SCSV [27,28]) in monetary units for the four construction periods, which is valuable for mainstreaming ES into policy and decision making; (3) To assess responses in SCSV change to LUCC by using a sensitivity indicator to identify those regions where a slight change in land use/cover could result in disproportionate changes of SCSV in the TGRA.

## 2. Study Area

The TGRA (28°56′–31°44′N, 106°16′–111°28′E), located upstream of the Yangtze River at the boundary of Chongqing Municipality and Hubei Province, encompasses an area of 59,900 km$^2$ and a human population of 16 million (Figure 1). It stretches narrowly along the Yangtze River from Jiangjin District of Chongqing Municipality to Yichang of Hubei Province, where the terrain is complex.

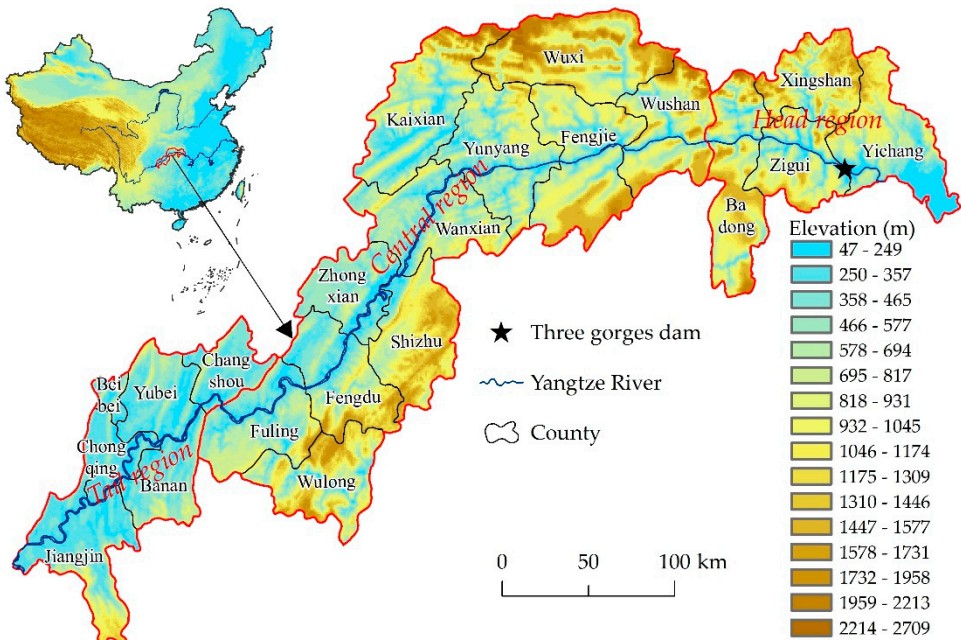

**Figure 1.** Location of the Three Gorges Reservoir Area (TGRA). The TGRA can be divided into three parts: head, central, and tail regions. The division between head and central regions is the provincial boundary of Hubei Province and Chongqing Municipality. The digital elevation model (DEM) was downloaded from Geospatial Data Cloud.

Mountainous areas comprise 74% of the TGRA, with just 4.3% of it plain area in the river valley and 21.7% composed of hilly area. Compared to the plains, mountain regions are ideal areas for exploring the impacts of LUCC on ES [20] because they are sensitive to human activities and natural disturbances are common there, namely landsides and mountainside collapses. The relationship between LUCC and ES seems more complicated in China's mountains because they harbor great population, and are undergoing dramatic industrialization and urbanization processes [20]. The TGRA is a typical mountain area that embodies this intensive, ongoing human–environment relationship. The TGRA's climate is subtropical monsoon, being located where the northern temperate zone shifts into the subtropical zone, where it is hot and rains much (annual rainfall of c. 1100 mm). Below 500 m, the TGRA valley has an annual temperature 17–19 °C with an annual frost-free period lasting 300–340 days. The soil here is dominated by purple (47.8%), yellow-brown soil (16.3%), and lime (34.1%) soil types [49].

## 3. Materials and Methods

### 3.1. Overview of Methodology

Figure 2 depicts the framework of this study. Firstly, the spatiotemporal characteristics of LUCC in the TGRA for 1995–2000, 2000–2005, 2005–2010, and 2010–2015 were analyzed by using several land use change indicators. Notably, these time periods corresponded to different construction phases of the TGP. Considering their division by the State Council Gorges Project Construction Committee Executive Office and by taking the relevant data availability into account, the whole study period was thus divided into the above four periods, which were distinguished by residents' first-time relocation and river interception (1995–2000), second relocation of residents and water level rising to 135 m (2000–2005), project completion and water level at 175 m (2005–2010), and TGP fully operational (2010–2015).

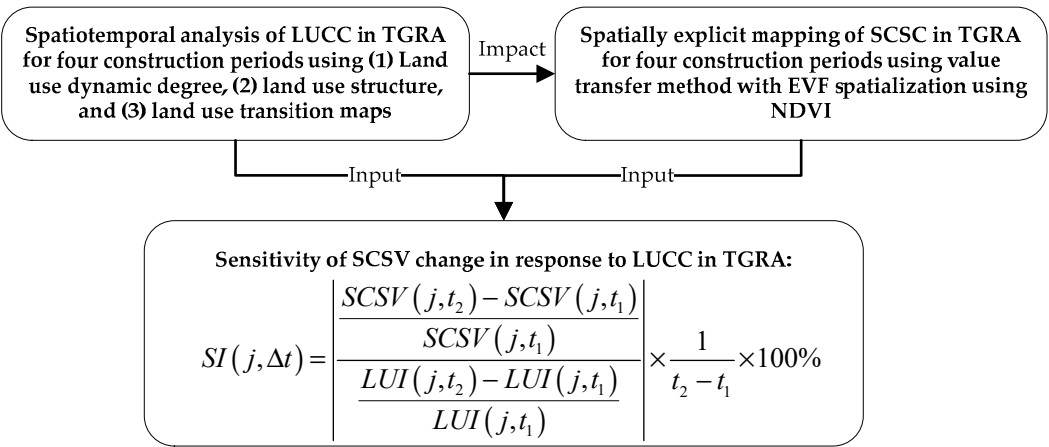

**Figure 2.** This study's framework. EVF: equivalent value factor; NDVI: Normalized Difference Vegetation Index; LUCC: land use/cover change; TGRA: Three Gorges Reservoir Area; SCSV: Soil Conservation Service Value; LUI: land use intensity.

Secondly, employing the value transfer method, responses of monetary SCSV to LUCC were estimated in a spatially explicit way. In particular, by taking the spatial heterogeneity of SCSV into account, the ecosystem-based equivalent value factor (EVF) table developed by Xie et al. [22] could be spatialized to 1 km according to NDVI, thus enabling further analyses of the SCSV's spatiotemporal characteristics Lastly, a sensitivity analysis of SCSV response to LUCC was conducted for regions of the TGRA.

### 3.2. Land Use/Cover Change Analysis

Land use dynamic degree (LUDD) and land use structure were relied upon to analyze the spatiotemporal characteristics of LUCC in the TGRA. LUDD can express the rate-of-change of a single land use type (single LUDD), or the overall state of land use rate-of-change (comprehensive LUDD). Land use structure and its change can reveal land use composition and the structural characteristics of land use change. Equations (1) and (2) are single and comprehensive LUDDs, respectively [50].

$$K_i = \frac{U_{i,t2} - U_{i,t1}}{U_{i,t1}} \times \frac{1}{\Delta t} \times 100\% \tag{1}$$

$$LC = \left[ \frac{\sum_i^n \Delta LU_{i-j}}{2\sum_{i=1}^n LU_i} \right] \times \frac{1}{T} \times 100\% \tag{2}$$

where $K_i$ is the LUDD of land use type $i$; $U_{i,t1}$ and $U_{i,t2}$ are the respective land use areas of land use type $i$ for the start and end of the study period; $\Delta t = t_2 - t_1$, the study period's duration; if $K_i > 0$, the land use type $i$ is in expansion for the period $\Delta t$ (otherwise, in contraction); $LC$ is the comprehensive LUDD for the study area, typically an administrative region; $LU_i$ denotes the initial area of land use type $i$ for the administrative region; $\Delta LU_{i-j}$ is the area of land use type $i$ converted to type $j$ over the study period for the administrative region; finally, $n$ is the number of land use types.

Corresponding spatial distribution maps of land use transitions for 1995–2000, 2000–2005, 2005–2010, and 2010–2015 were drawn to explore trends in land use change in TGRA. Analyses were performed for the entire study period (i.e., 1995–2015) and separately for the four TGP construction periods.

### 3.3. Estimating Soil Conservation Service Value Due to Land Use/Cover Change

The value transfer method is often useful to decision makers who must assess the ES values at regional or national scales, especially when primary data at these scales is lacking; its popularity stems from its quick assessment and low cost of collecting primary data [11,14]. In this study, we specifically adopted the value transfer method as developed by Costanza et al. [11] and Xie et al. [22]. In it, they had estimated the economic value of food production service for per unit area farmland ecosystem, which is usually simple to measure, calling it the economic value of one standard EVF. Then, based on a survey of more than 700 ecological experts, Xie et al. [22] developed an EVF table in which the relative weight of one ES for a certain ecosystem could be compared with food production service for a farmland ecosystem. Finally, the value of each ES for each ecosystem was estimated as the product of the economic value of one standard EVF, an EVF (dimensionless), and the geographical area of that ecosystem. This method has been widely used, especially for understanding ESV changes due to LUCC [12,20], and was used in the current study to estimate SCSV due to LUCC for the TGRA.

The ES value for one ecosystem type spanning multiple regions will differ among them because of spatial heterogeneity in vegetation cover, biomass, and productivity within that ecosystem type [13]. Therefore, taking the spatial heterogeneity of a given ecosystem into account, the EVF was first spatialized before the SCSV estimation. Studies indicate a strong positive correlation between NDVI and SCSV [51,52], so NDVI was used to spatialize the EVF using Equations (3) and (4).

$$f(i,j,t) = \frac{NDVI(i,j,t) - \min(NDVI(i,j,t))}{\max(NDVI(i,j,t)) - \min(NDVI(i,j,t))} \tag{3}$$

$$E_f(j,t) = \frac{f(i,j,t)}{f(i,t)} \times E(i,t), i = 1, 2, \cdots, n \tag{4}$$

where $NDVI(i,j,t)$ is normalized difference vegetation index of grid $j$ within ecosystem type $i$ for year $t$; $\min(NDVI(i,j,t))$ and $\max(NDVI(i,j,t))$ are respectively the minimum and maximum of $NDVI(i,j,t)$; $f(i,j,t)$ is the normalized $NDVI(i,j,t)$, and $f(i,t)$ the mean NDVI for all grids within ecosystem type $i$ for

year *t*; $E_f(j,t)$ is the spatialized EVF for the SCSV of grid *j* for year *t*, and; $E(i,t)$ is EVF for the SCSV of ecosystem type *i* for year *t*.

The economic value of one standard EVF is one-seventh the market value of the average grain yield per unit area [22,53]. The average grain yield per unit area for the TGRA is 468,350 kg·km$^{-2}$ with an average grain price of 0.23 dollars per kilogram (USD·kg$^{-1}$) for 1995, 2000, 2005, 2010, and 2015 [54,55]. To avoid the effects of inflation, we used the average grain price from 1995 to 2015. The economic value of one standard EVF for the TGRA is thus 15,690 USD·km$^{-2}$. Equations (5) and (6) were then used to calculate the SCSVs in the five years (1995, 2000, 2005, 2010, and 2015).

$$SCSV(j,t) = A \times V \times E_f(j,t) \tag{5}$$

$$SCSV(t) = \sum_{j=1}^{n} SCSV(j,t) \tag{6}$$

where $SCSV(j,t)$ is the soil conservation service value of grid *j* in year *t*; *A* is grid area (=1 km$^2$); *V* is the economic value of one standard EVF (= 15,690 USD·hm$^{-2}$); $SCSV(t)$ is the total *SCSV* for the study area for year *t*.

Spatiotemporal characteristics of SCSV and their changes for 1995–2000, 2000–2005, 2005–2010, and 2010–2015 were analyzed from the perspective of varying altitude and slope to reveal the vertical spatial heterogeneity in SCSV.

### 3.4. Sensitivity Analysis of Soil Conservation Service Value Changes to Land Use/Cover Change

To identify differential responses of SCSV to LUCC, a sensitivity analysis was also conducted. Following Song et al. [14,16], a modified sensitivity index expressed in Equation (7) was developed to determine SCSV responses to land use change in a spatially explicit way.

$$SI(j, \Delta t) = \left| \frac{\frac{SCSV(j,t_2) - SCSV(j,t_1)}{SCSV(j,t_1)}}{\frac{LUI(j,t_2) - LUI(j,t_1)}{LUI(j,t_1)}} \right| \times \frac{1}{\Delta t} \times 100\% \tag{7}$$

where $SI(j,\Delta t)$ is the sensitivity index of grid *j* for the period $\Delta t = t_2 - t_1$ (unit: %); $SCSV(j,t_1)$ and $SCSV(j,t_2)$ are respectively the soil conservation service values of grid *j* for years $t_1$ and $t_2$ (unit: USD); $LUI(j,t_1)$ and $LUI(j,t_2)$ are the land use intensities of grid *j* for years $t_1$ and $t_2$ (dimensionless). Compared with the indicator proposed by Song et al. [14,16], the LUCC used changes to grid cell-based land use intensity [56] instead of relying upon an administrative division-based land use dynamic degree; this should provide more spatial information in quantity. The human influence scores assigned to each type are listed in Table 1. The higher the sensitivity index, the more the SCSV is sensitive to LUCC in this grid cell; these areas would likely require additional attention and protection.

**Table 1.** Land use types [57,58] and the human influence scores assigned to each type based on the relative disturbance degree calculated in relevant studies [31,59–61]. This table was taken from reference [56] and revised further.

| 1st Level Classes | 2nd Level Classes | Descriptions | Score |
|---|---|---|---|
| Construction land | Urban built-up | Lands used for urban | 10 |
| | Rural settlements | Lands used for settlements in villages | 8 |
| | Others | Lands used for factories, quarries, mining, oil-field slattern outside cities, and for special uses, such as transportation and airports | 9 |
| Cropland | Paddy/dry land | Cultivated lands for crops, such as mature cultivated land, newly cultivated land, fallow, and shifting cultivated land; intercropping land, such as crop-fruiter, crop-mulberry, and crop-forest land in which a crop is a dominant species; bottomland and beach areas cultivated for ≥3 years | 7 |
| Grassland | Dense grassland | Grassland with canopy coverage >50% | 2 |
| | Moderate grassland | Grassland with canopy coverage between 20% and 50% | 1 |
| | Sparse grassland | Grassland with canopy cover between 5% and 20% | 0 |
| Forestland | Forest | Natural or planted forests with canopy cover >30% | 1 |
| | Shrub | Lands covered by trees <2 m high with the canopy cover >40% | 0 |
| | Woods | Lands covered by trees with canopy cover between 10–30% | 0 |
| | Others | Lands such as tea-garden, orchards, groves, and nurseries | 2 |
| Water body | Stream and rivers | Lands covered by rivers, including canals | 1 |
| | Reservoir/ponds | Man-made facilities for water reservation | 1 |
| | Bottomland | Lands between normal water levels and flood levels | 1 |
| | Lakes | Lands covered by lakes | 0 |
| Barren land | See descriptions | Lands not put into practical use or simply difficult to use, such as sandy land, Gobi, salina, swampland, bare soil, and bare rock, among others. | 0 |

### 3.5. Materials and Pre-Processing

Remotely sensed LULC (land use/cover) data of the TGRA for 1995, 2000, 2005, 2010, and 2015 were taken from China's Land-Use/cover Datasets (CLUDs) [58,62]. Based on Landsat TM/ETM+ data and using a human–computer interactive interpretation method, Liu et al. [58] reconstructed CLUDs with six classes of land use/cover type (i.e., cropland, forestland, grassland, water body, unused land, and built-up land) and 25 subclasses of land use/cover type, which were updated regularly at 5-year intervals from the late 1970s to 2015. These CLUDs, with a resolution of 1 km, can be downloaded from the Resource and Environment Data Cloud Platform (http://www.resdc.cn/). The overall classification accuracy of CLUDs for the six classes of land use/cover type reaches 94.3% while for the 25 subclasses type it reaches 91.2% [58,62]; hence, both satisfy the requirement of user mapping accuracy at the 1:100,000 scale. The NDVI datasets for TGRA for 1998, 2000, 2005, 2010, and 2015, were downloaded from the Resource and Environmental Data Cloud Platform, and used to revise the equivalent value factor. Based on SPOT/VEGETATION NDVI data and using a maximum value composite method, Xu et al. [63] developed the annualized 1 km NDVI spatial distribution dataset for China dating back to 1998.

Additionally, statistics from the Yearbooks for Chongqing Municipality and Hubei Province [54,55] were used to calculate the ES value of the standard equivalent value factor (EVF). The 90-m digital elevation model (DEM) which was produced based on SRTM3 Version 4.1 was downloaded from the Geospatial Data Cloud (http://www.gscloud.cn/), then resampled to 1 km for use in clarifying the vertical spatial heterogeneity of the SCSVs.

## 4. Results

### 4.1. Analysis of Land Use/Cover Change for 1995–2015

The area of forestland was largest among the six land use/cover types for the TGRA, accounting for about 47% of the total land area, and is mainly distributed in mid-eastern regions of the TGRA, Fuling County, Wulong County, and the southern part of Jiangjin County (Figure 3). The percentage of cropland in the TGRA was approximately 38%, mainly distributed in mid-western regions of the TGRA. In third place was grassland, representing 12.7% of the TGRA's area.

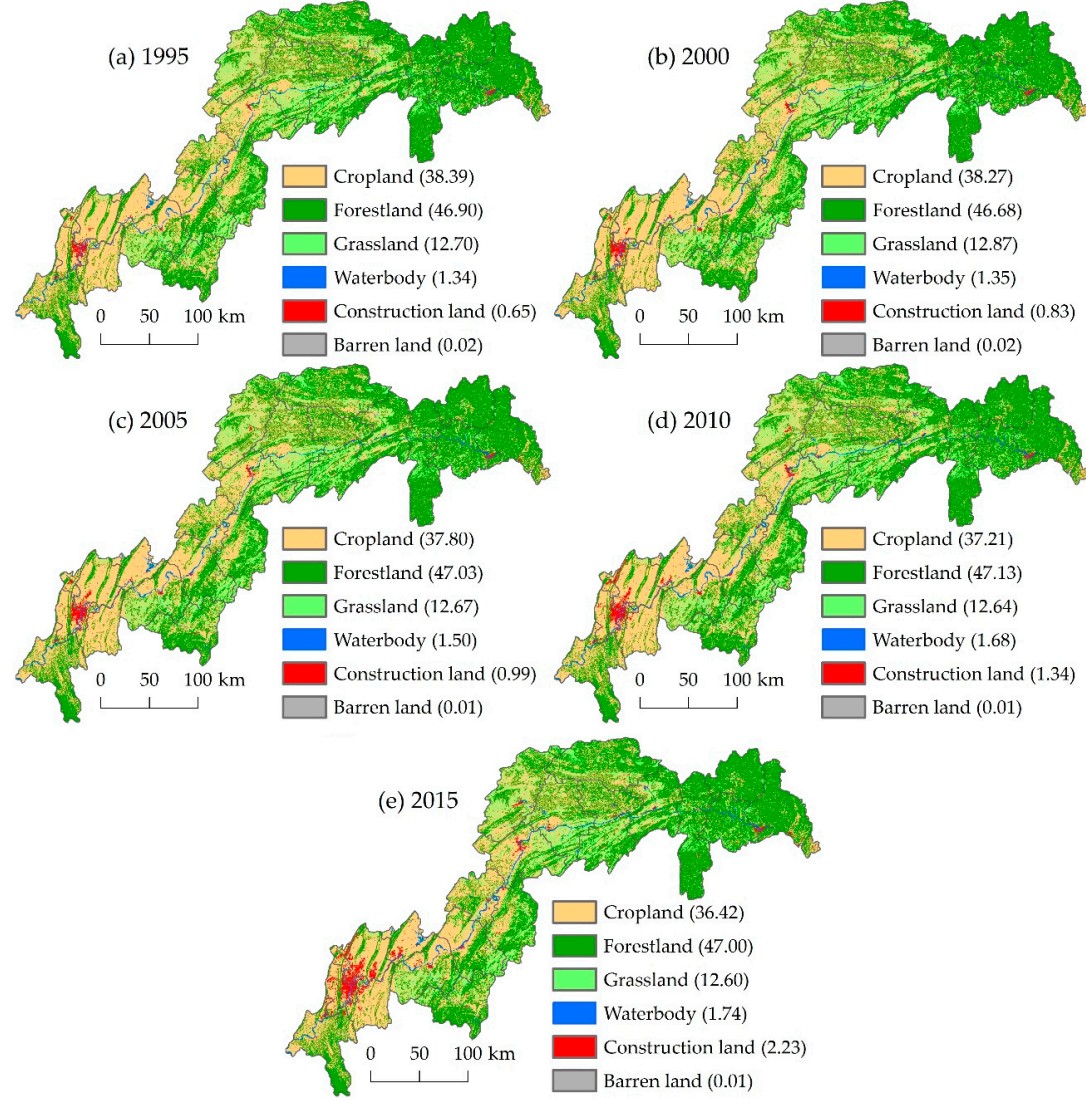

**Figure 3.** Land use/cover maps in the Three Gorges Reservoir Area for (**a**) 1995, (**b**) 2000, (**c**) 2005, (**d**) 2010, and (**e**) 2015. The data is sourced from references [58,62], which can be downloaded from the Resource and Environment Data Cloud Platform. The number in parentheses is the area percentage of each land use type.

Although the areas corresponding to construction land and water body are small, they have been increasing in 1995–2015 (Figure 3), respectively by 910.7 km$^2$ (242.92%) and 227.7 km$^2$ (29.45%). The single LUDDs of construction land and water body were 12.12% and 1.47%, respectively, while it was 0.26% for cropland, which decreased by 1134.4 km$^2$ (5.13%) over the two decades. Cropland, forestland, and grassland had small single LUDDs due to their large initial areas, especially the forestland is only 0.01% (Table 2). The comprehensive LUDD of the TGRA for the entire 20-year period was 0.11%.

**Table 2.** Area of land use change and land use dynamic degree for the Three Gorges Reservoir Area during four dam construction periods, and for the entire 20-year period.

| Land Use Types | 1995–2000 | | 2000–2005 | | 2005–2010 | | 2010–2015 | | 1995–2015 | |
|---|---|---|---|---|---|---|---|---|---|---|
| | AOC [a] (km$^2$) | LUDD [b] (%) | AOC (km$^2$) | LUDD (%) | AOC (km$^2$) | LUDD (%) | AOC (km$^2$) | LUDD (%) | AOC (km$^2$) | LUDD (%) |
| Construction land | 102.1 | 5.41 | 90.5 | 3.77 | 203.1 | 7.13 | 515.1 | 13.3 | 910.7 | 12.12 |
| Water body | 1.7 | 0.04 | 87.5 | 2.26 | 104.6 | 2.42 | 34.0 | 0.7 | 227.7 | 1.47 |
| Forestland | −128.2 | −0.09 | 204.7 | 0.15 | 54.0 | 0.04 | −72.6 | −0.1 | 57.8 | 0.01 |
| Cropland | −70.3 | −0.06 | −269.4 | −0.24 | −339.7 | −0.31 | −455.1 | −0.4 | −1134.4 | −0.26 |
| Grassland | 94.8 | 0.26 | −110.8 | −0.30 | −21.1 | −0.06 | −21.4 | −0.1 | −58.5 | −0.04 |
| Barren land | 0 | 0.00 | −2.5 | −5.13 | −0.6 | −1.62 | 0 | 0.0 | −3.1 | −1.58 |

[a]. AOC: area of change. [b]. LUDD: single land use dynamic degree.

## 4.2. Analysis of Land Use/Cover Change for Different Construction Periods

The LUCC characteristics of TGRA under the four construction periods of TGP were next explored. For 1995–2000, the comprehensive LUDD was 0.07%. Construction land increased by 102.1 km$^2$, because of relocating residents, villages, and towns for the first time, and it was mainly distributed in Chongqing, Wanxian County, and regions along the Yangtze River (Figure 4a). Interception of the Yangtze River and initial water storing by the reservoir in 1997 increased the area of water body slightly in this period, with a single LUDD of 0.04% (Table 2). In addition, the conversion of forestland to grassland was obvious in Wuxi County for this period.

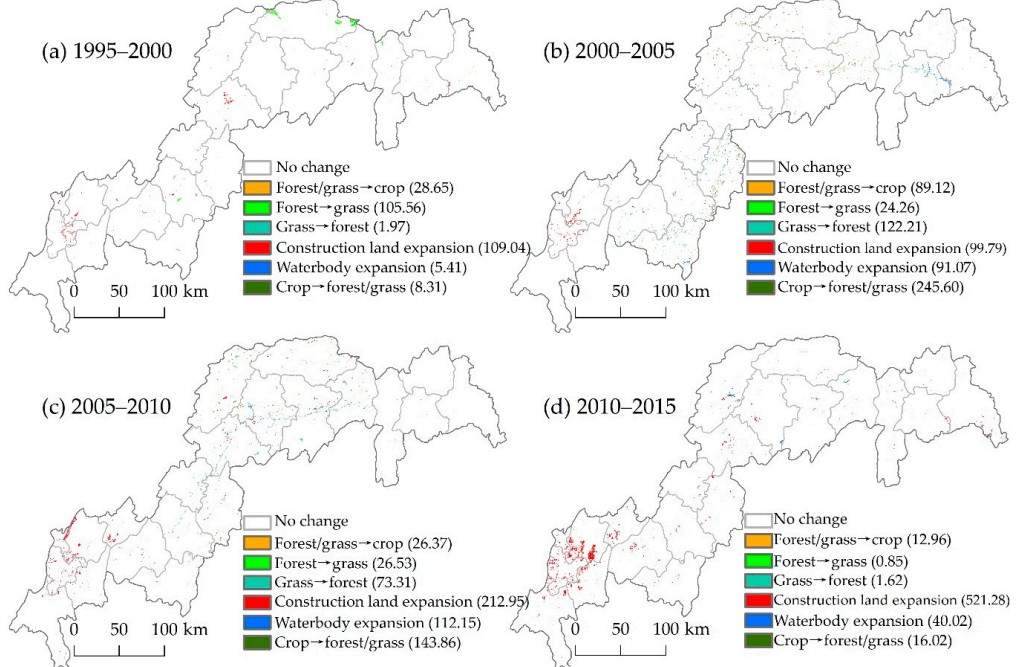

**Figure 4.** Land use transformation in the Three Gorges Reservoir Area for (**a**) 1995–2000, (**b**) 2000–2005, (**c**) 2005–2010, and (**d**) 2010–2015. The numbers in parentheses indicate the total area (unit: km$^2$) transformed.

For 2000–2005, the second relocation of residents began and reservoir's water level rose to 135 m. As a result, both construction land and water body continued to increase in area. The former's expansion happened not only in Chongqing but also in the Yubei district, while the area of water body increased by 91.1 km$^2$ with a single LUDD of 2.26% (Table 2), largely because of water storage by the reservoir, distributed predominantly in the head part of the TGRA, which includes Zigui, Badong, and Wushan Counties (Figure 4b). Furthermore, in 2002 the Grain for Green Program was implemented in the TGRA, thus driving the conversion of cropland to forestland or grassland, which amounted to 245.6 km$^2$ for this period. This conversion mainly occurred in Shizhu, Wulong, Fuling, and Fengdu Counties (Figure 4b). Comparatively, during 1995–2000, this conversion totaled only 8.3 km$^2$.

For 2005–2010, its main characteristics were dam project completion and a reservoir water level that now reached 175 m. In this period, construction land area increased by 203.1 km$^2$, mainly distributed in Chongqing, Yubei, Beibei, and Changshou districts of the tail part of the reservoir, as well as Wanxian, Yunyang, Fengjie, and Wushan Counties of the central part of reservoir (Figure 4c). The water body area increased by 104.6 km$^2$, primarily in Fengdu and Wushan Counties of the reservoir's central part, while 143.9 km$^2$ of cropland was converted to forestland/grassland as the Grain for Green Program continued its implementation. However, the total cropland area decreased by 269.4 km$^2$ in 2005–2010 because of construction land and water body expansion, in addition to the Grain for Green Program.

The final period, 2010–2015, featured a fully operational TGP, for which the occupation of cropland for construction land was a significant feature. Because of economic development and supporting facilities built for the dam, the construction land area grew by 515.1 km$^2$ with a single LUDD of 13.33% (Table 2), being distributed in Chongqing and its surrounding counties, particularly the Yubei district. The cities or towns along the Yangtze River also underwent expansion of construction land during this period (Figure 4d). Conversely, cropland area decreased by 455.1 km$^2$, with a single LUDD of –0.42%, the maximum among the four periods. Since the dam's completion, the water body area increased slowly, with a single LUDD of 0.70%. Further, the transformation of cropland to forestland/grassland was small during this period, as the Grain for Green Program had ended in 2007.

In sum, the expansion of construction land and water body and contraction of cropland are the main characteristics of LUCC in the four periods.

*4.3. Soil Conservation Service Value under the Influence of Land Use/Cover Change*

The SCSV for the TGRA was generally stable, slightly reduced from 2.447 billion USD in 1995 to 2.424 billion USD in 2015, a decrease of <1%. The spatial distributions of SCSV for the five years are illustrated in Figure 5. It can be seen that the SCSV for the head part and several counties (including Shizhu, Fengdu, and Wulong) of the central and tail parts of the TGRA is high, and the SCSV for the mid-west of the TGRA is low, especially for Chongqing and its surrounding regions (Figure 5). The SCSV for regions along the Yangtze River is also low.

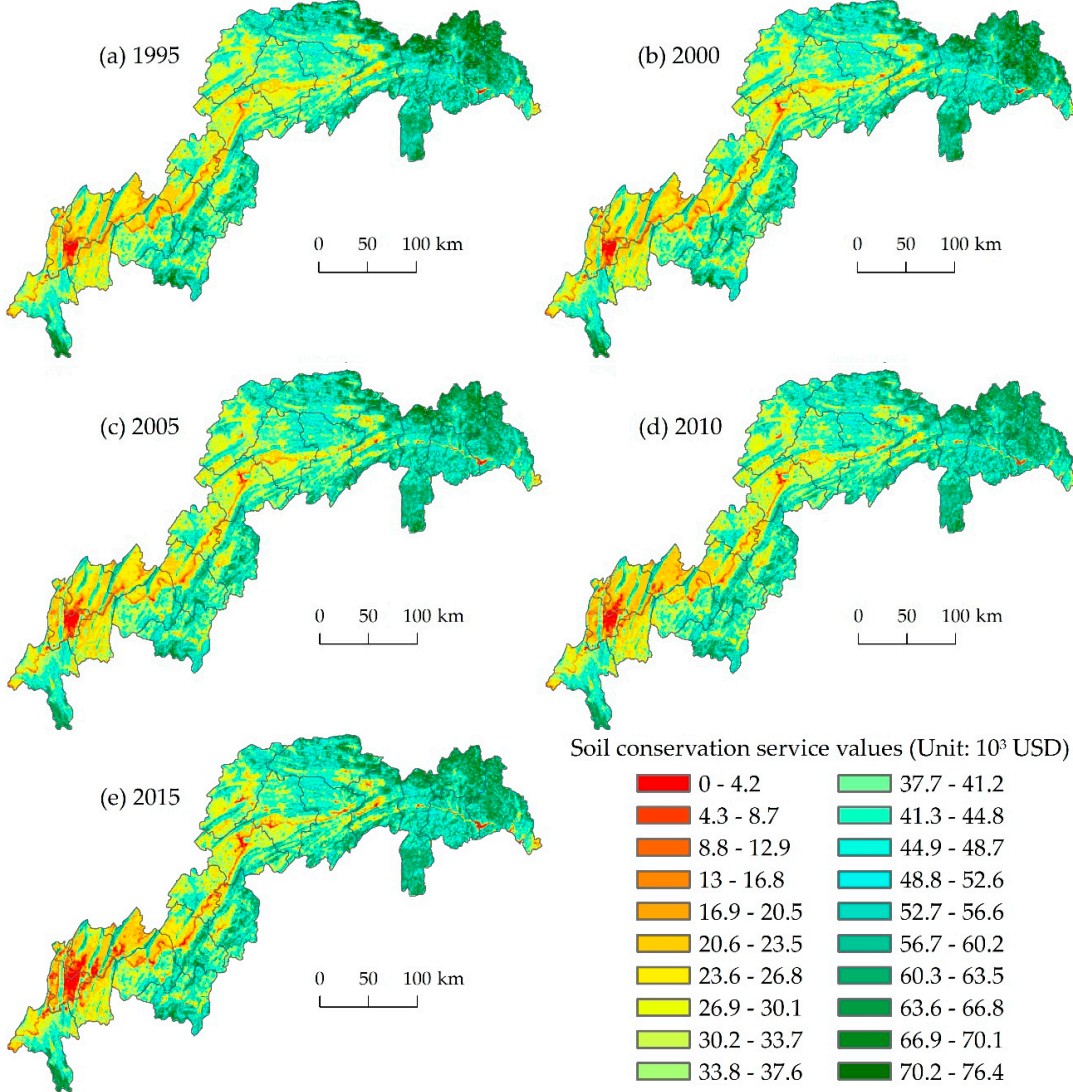

**Figure 5.** Soil conservation service values (SCSV) of the Three Gorges Reservoir Area for (**a**) 1995, (**b**) 2000, (**c**) 2005, (**d**) 2010, and (**e**) 2015; their respective total SCSVs are 2.447, 2.441, 2.444, 2.440, and 2.424 billion USD.

Vertical spatial heterogeneity in SCSV was also investigated (Figure 6). In terms of elevation, we followed a 100-m interval for statistics; clearly the SCSV followed a bimodal curve as a function of increasing elevation (Figure 6a). Below 200 m, the SCSV is small but it increased rapidly with higher elevation, peaking first at 400–1000 m, then steeply declining to 1400 m, where it rose again until 1600 m, its approximate second peak. Beyond that, the SCSV decreased rapidly with increasing altitude. As such, the proportion of SCSV found at >2000 m elevation is very small.

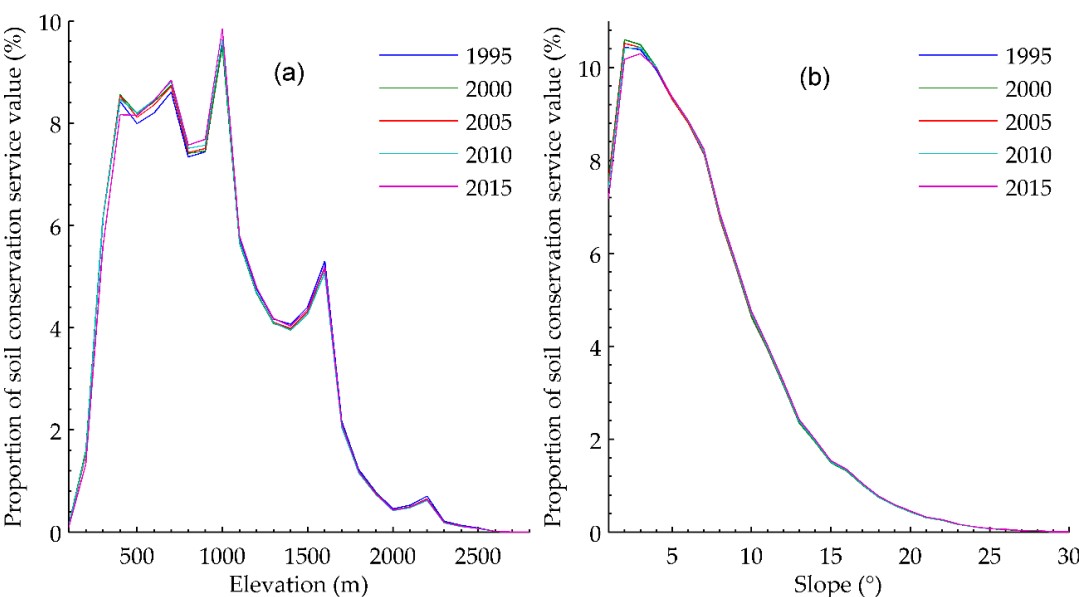

**Figure 6.** Vertical spatial heterogeneity of the soil conservation service value (SCSV). The SCSV percentage change along (**a**) elevation and (**b**) slope in the Three Gorges Reservoir Area.

With respect to slope, a one-degree interval was followed for statistics. In this case, the SCSV had a unimodal curve as a function of greater slope (Figure 6b), increasing quickly in the 0–3° interval, after which it decreased rapidly. Evidently, the SCSV was mainly concentrated in those areas with small slopes: in the 0–10° range, its proportion of all values exceeded 80%.

Over the study period, SCSV's vertical spatial heterogeneity remained stable. Only a slight fluctuation of SCSV at elevations of 400–900 m and the 2–3° slope were detected for the five years (Figure 6).

Next we mapped relative changes in the spatial pattern of SCSV for the four periods (Figure 7). For 1995–2000, the SCSV decreased in eastern TGRA—including Xingshan, Zigui, Badong, Wushan, and Wuxi Counties, Wulong County, and Chongqing—whereas it increased in western TGRA (Figure 7a). Further, the SCSV for regions along the main stream of the Yangtze River had also decreased. For 2000–2005, this spatial pattern was reversed when compared with the prior period (Figure 7b), yet a significant decrease of SCSV in Chongqing and Yubei County persisted. The reduced SCSV along the main stream of the Yangtze River in the head part of the TGRA could also be gleaned. Compared with two periods before, the areas where SCSV increased and decreased are more or less evenly distributed for 2005–2010 (Figure 7c), with a declining SCSV shifting from Chongqing to the Yubei and Changshou Counties. The regions along the main stream of the Yangtze River in the central part of the TGRA experienced SCSV increases for 2005–2010. For the last period, 2010–2015, SCSV declined considerably in the surrounding regions of Chongqing, which likely contributed to its expanded construction land during this period (Figure 7d). The SCSV decrease was also significant for the newly constructed urban areas along the Yangtze River during 2010–2015.

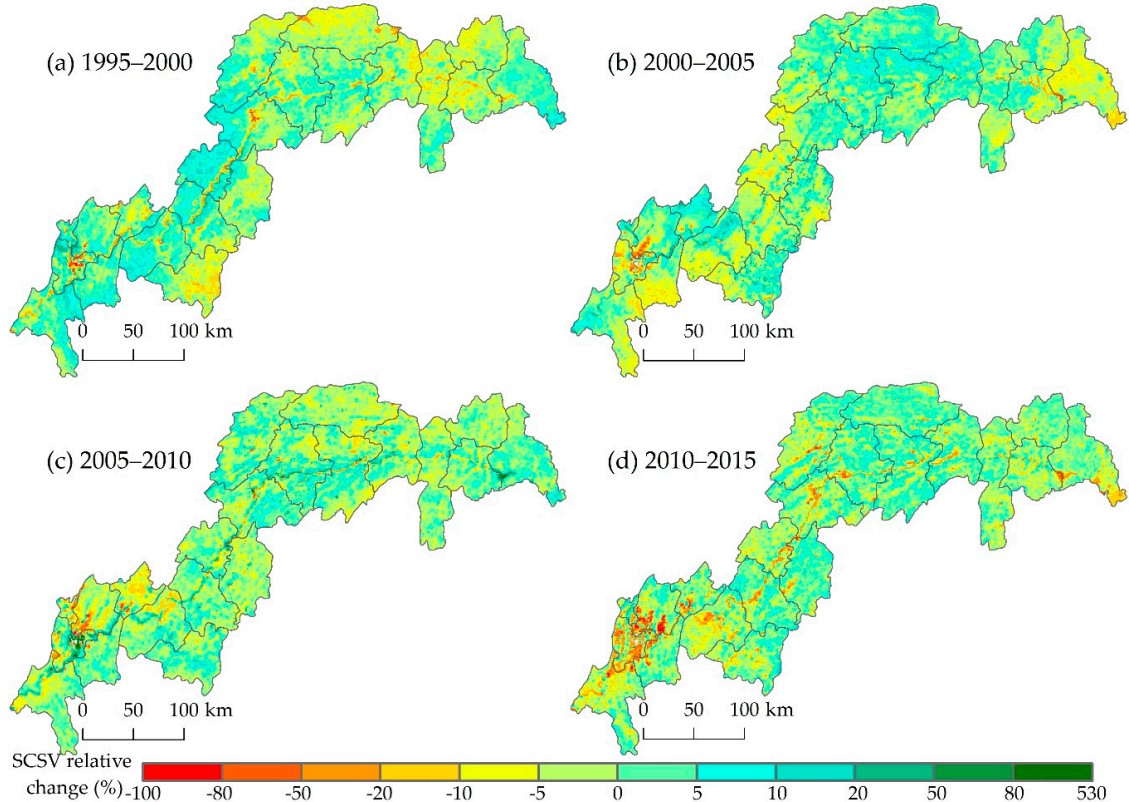

**Figure 7.** Changes of soil conservation service values in the Three Gorges Reservoir Area for (**a**) 1995–2000, (**b**) 2000–2005, (**c**) 2005–2010, and (**d**) 2010–2015.

*4.4. Sensitivity of Changes of Soil Conservation Service Value to Land Use/Cover Change*

By combining the data on LUCC and SCSV changes, the sensitivity index of SCSV change to LUCC in the TGRA was obtained, which averaged 1.04, 0.53, 0.92, and 1.25 for 1995–2000, 2000–2005, 2005–2010, and 2010–2015, respectively. Therefore, for every 1% change in land use intensity, the SCSV fluctuated by 1.04%, 0.53%, 0.92%, and 1.25% (on average) in these four periods. Evidently, this sensitivity decreased at first but then increased continuously.

Figure 8 shows the spatial patterns of sensitivity index of SCSV changes to LUCC. The sensitivity index in Chongqing and its surrounding regions was large, expanding gradually from 1995–2000 to 2010–2015, which is the most obvious spatial variation characteristic of the sensitivity index for the four periods. Viewed from the different periods, the sensitive area expanded in the south of Wulong, Fengdu, and Shizhu Counties, and the head-central parts of the TGRA, from 1995–2000 to 2000–2005, yet the sensitivity index values for this newly expanded regions are low. From 2000–2005 to 2005–2010, the sensitive area expanded in the central part of the TGRA and in those regions along the main stream of the Yangtze River; but again, their sensitivity index values were low. Compared with 2005–2010, the sensitive area shrank in the middle part of the TGRA yet expanded slightly in the head part of the TGRA during 2010–2015. Additionally, regions with a high sensitivity index for 2010–2015 expanded not only in Chongqing and its surrounding regions, but also in some new urban areas along the main stream of the Yangtze River.

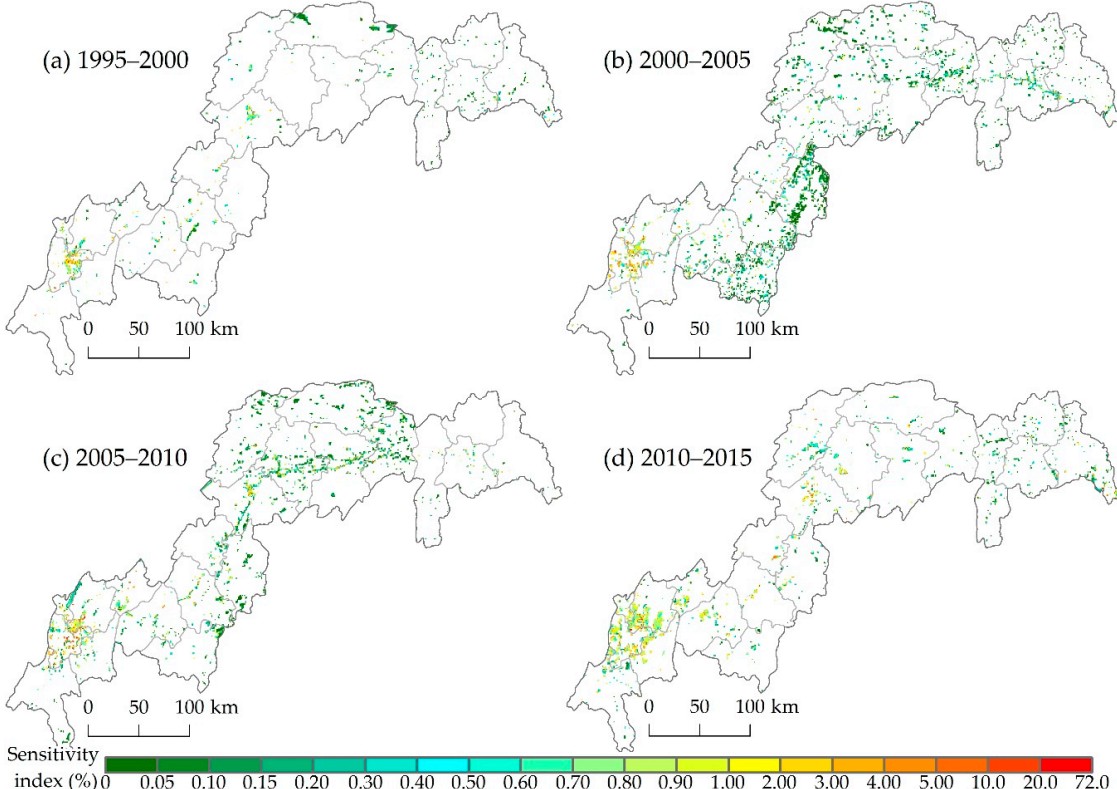

**Figure 8.** Spatial patterns of the sensitivity index of soil conservation service value (SCSV) changes in response to land use/cover change (LUCC) in Three Gorges Reservoir Area, for the periods (**a**) 1995–2000, (**b**) 2000–2005, (**c**) 2005–2010, and (**d**) 2010–2015. No LUCC occurred in the blank (white) areas.

An averaged sensitivity index of SCSV change to LUCC was also calculated per county for the four periods, to facilitate its use by local governments (Table 3). It can be seen that Chongqing and its surrounding Counties—including Yubei, Beibei, Banan, Jiangjin, and Changshou—had high sensitivities, as did Zhongxian County. As Figure 5 reveals, the SCSVs for these regions are generally low, and their respective declines in SCSVs are evident over the entire study period.

**Table 3.** Averaged sensitivity index of SCSV (soil conservation service value) change in response to land use/cover change in each county of the Three Gorges Reservoir Area for the four construction periods. The green and red points in the trend line denote the minimum and maximum sensitivities, respectively.

| Classification | County | 1995–2000 | 2000–2005 | 2005–2010 | 2010–2015 | Trendline | Average |
|---|---|---|---|---|---|---|---|
| Increase continuously | Yichang | 0.26 | 0.6 | 0.79 | 0.88 | | 0.63 |
| | Changshou | 0.81 | 0.92 | 1.36 | 1.61 | | 1.18 |
| | Yunyang | 0.28 | 0.33 | 0.62 | 1.52 | | 0.69 |
| Increase with fluctuations | Zhongxian | 1.63 | 0.8 | 0.56 | 2.97 | | 1.49 |
| | Xingshan | 0.22 | 0.1 | 0.01 | 0.27 | | 0.15 |
| | Wulong | 0.63 | 0.24 | 0.31 | 0.79 | | 0.49 |
| | Fuling | 1.21 | 0.66 | 0.66 | 1.23 | | 0.94 |
| | Wanxian | 1.17 | 0.37 | 0.76 | 1.54 | | 0.96 |
| | Fengdu | 1.05 | 0.32 | 0.62 | 1.3 | | 0.82 |
| | Kaixian | 0.36 | 0.11 | 0.3 | 1.18 | | 0.49 |
| | Wuxi | 0.2 | 0.16 | 0.16 | 0.51 | | 0.26 |
| | Zigui | 0.23 | 0.66 | 0.93 | 0.41 | | 0.56 |
| | Badong | 0.22 | 0.28 | 0.57 | 0.34 | | 0.35 |
| | Banan | 0.79 | 1.64 | 0.85 | 1.55 | | 1.21 |
| Decrease with fluctuations | Chongqing | 2.59 | 2.55 | 3.94 | 1.72 | | 2.70 |
| | Yubei | 1.83 | 1.99 | 2.63 | 1.78 | | 2.06 |
| | Beibei | 1.71 | 1.6 | 0.9 | 1.27 | | 1.37 |
| | Shizhu | 1.17 | 0.18 | 0.25 | 0.82 | | 0.61 |
| | Fengjie | 2.45 | 0.2 | 0.65 | 0.61 | | 0.98 |
| | Wushan | 0.49 | 0.37 | 0.39 | 0.39 | | 0.41 |
| Decrease continuously | Jiangjin | 1.57 | 1.12 | 1.09 | 1.07 | | 1.21 |

Considering the trend in sensitivity index changes in the four periods, all counties (21 in total) were divided into four categories: continuous increase, increase with fluctuations, decrease with fluctuations, and continuous decrease (Table 3). The number of counties whose sensitivity index increased over the 20-year period amounted to 14, accounting for two-thirds of the total and 70.63% of the entire study area. Actually, three of the 14 increased continuously, and 11 of the 14 increased with fluctuations. Most of them were in the head-central parts of the TGRA where the SCSVs for these counties were relatively high. Protecting the SCSV in these regions should be sought after. Seven counties had lowered sensitivity index over the study period, accounting for one-third of the total, with the sensitivity index of Jiangjin County undergoing a continuous decrease. Although their sensitivity index values decreased, policymakers should nonetheless pay sufficient attention to them since the absolute sensitivity of them is high, and these counties feature substantial urbanization and must support dense populations.

## 5. Discussion

### 5.1. Characteristics of Land Use/Cover Change in the TGRA for Different Periods

The main construction projects in each phase of the TGP were different, consequently their associated LUCC characteristics and driving factors likely differed as well. So, in this study we not only performed the LUCC analysis for the entire study period (1995–2015), but also for each 5-year construction period of the TGP, which is unavailable in most previous studies [35,36,48]. With such period-specific results, we obtained the characteristics and driving forces of LUCC in the TGRA unique to the four construction periods, which strengthens its practical use by decision makers. We found

that single and comprehensive LUDDs and the main driving factors of LUCC were not alike in the different periods.

For 1995–2000, relocation activity resulted in the expansion of construction land in regions along the main stream of the Yangtze River. Hence, the formulation and implementation of land use policies related to immigration should be the focus of government work at this stage. For 2000–2005, water storage in the dam's reservoir was responsible for water body expansion in the head part of the TGRA, while the Grain for Green Program spurred conversion of cropland to forestland/grassland. Continued water storage by the reservoir and the further implementation of the Grain for Green Program were the main driving forces of water body expansion and reforestation during 2005–2010; when compared with 2000–2005, these shifted to central and central-north parts of the TGRA, respectively. Assessing the ecological effects of water body expansion along the Yangtze River [25,26] and the effectiveness of the Grain for Green Program for 2000–2010 should be a priority for scholars and governments. For the final period, 2010–2015, cropland displacement by construction land was the outstanding characteristic because of rapid urbanization in the surrounding regions of Chongqing and along the main streams of the Yangtze River. Reasonable planning of urbanization [30] projects and protecting cropland for food security must be the focus of this period.

*5.2. Comparisons with Previous Studies of Soil Conservation Service Value Mapping*

The direction and magnitude of change seen in total SCSV for 2000–2015 revealed in this study agrees with findings from Wu et al. [44] for 2000–2015 for the Chongqing section of the TGRA. Similarly, our results for vertical spatial heterogeneity in the monetary SCSV are also consistent with those of reference [46], which was a spatially explicit assessment of SCSV in biophysical terms using the InVEST model [9]. These general agreements support the reliability of our results.

Spatially explicit valuation of ES in monetary units ought to reveal more details of ES's spatial patterning. Previous studies of LUCC related to SCSV change were mostly done at the administration polygon level [12,14,16,20,44], which can only provide suggestions for analyses of ES governance at broad scales. Although the outputs of assessment based on process-based ecosystem models are in the form of grid cell-based maps [10,46], this is predicated upon tremendous primary data inputs and collecting this needed data is often resource intensive, thus rendering this approach limited in data-scarce areas. In this study, we spatialized the EVF using NDVI and improved the valuation scale to a 1-km resolution, all at a low cost, given its easy implementation. The "remotely sensed" SCSV is more valuable than the polygon-based SCSV for ES governance and sustainable development planning since the policies needed by different areas likely differ because of varying natural and social conditions. In addition, the spatially explicit SCSV database is also more valuable in related downstream studies. For example, it can be used to assess improvements to ES from financial investments in natural capital [3,64] and the effectiveness of protected areas for ES [65].

The TGP brought with it huge economic benefits in many aspects, including flood control, power generation, and shipping. Further, the total SCSV for TGRA was generally stable for the whole study period, thus indicating the human–environment relationship is generally harmonious and human activities had little adverse impacts on SCSV in the TGRA during 1995–2015. This sustainable situation may have resulted from the implementation of ecological engineering, including the Grain for Green Program and transforming the sloping cropland into terraces in hilly and mountainous areas. Nonetheless, there are still opportunities for conservation gains. We should pay close attention to SCSV losses in Chongqing and its surrounding regions and those regions along the Yangtze River because of their prevalent urbanization and relocated villages and towns.

*5.3. Spatially Explicit Identification of Soil Conservation Service Value Sensitive Areas in Response to Land Use/Cover Change and Protection Policies for Them*

By using land use intensity to denote LUCC, we revised the sensitivity index of previous studies [14,16,20]. This revised sensitivity index can determine the SCSV-sensitive areas in response to

LUCC at the grid cell scale, which should provide more robust information on processes of landscape and ecological planning by delineating the boundaries of these sensitive areas more accurately. We also calculated an averaged sensitivity index for each county in each of the four periods, to help local government better deal with this sensitivity data and their implications.

Our sensitivity results let us identify those SCSV areas most sensitive to human land use activities in the TGRA for 1995–2015. We suggest that more attention should be paid to these regions by policy makers to prevent SCSV losses. The high sensitivity index for Chongqing and its surrounding regions is best explained by the occupation of cropland by rapid urbanization. The ecological protection policies for this area should thus target the protection of cropland. For instance, greater fees could be imposed for converting cropland into construction areas, while permanent cropland, whose use would not change under any circumstances, must be designated as such to avoid encroachment from new building. In addition, land consolidation in rural settlements should be encouraged [14]. For those 14 counties with high SCSVs and a low but increasing sensitivity, protection of their SCSV by setting up protected areas for SCSV should be pursued, as likewise suggested by Xu et al. [65] based on their national-scale analysis.

*5.4. Limitations and Caveats*

This study's results do contain some uncertainties that should be heeded by planners, as well as scholars, aiming to improve the valuation of ES in the TGRA. First, we suggest they focus on the change trends and not the absolute value of the reported SCSVs. The latter was calculated based on the grain yield and price, which is affected by multiple factors, including improved agriculture technology, climate change, and present-day market situations. To overcome this, an average value for the whole study period was used but even so it only captures the overall level of SCSVs, not their true values. In addition, the dynamic mechanism of LUCC as it affects SCSV was not analyzed. LUCC first influences specific ecological processes, including hydrological processes and soil erosion, before they can have an impact on ES. In future studies, the ecological processes and mechanisms of LUCC affecting ES should be rigorously explored.

The sensitivity of SCSV in response to LUCC for regions without detectable land use/cover change cannot be explored by our sensitivity index (the blanks in Figure 8), which is another limitation of this work and the original elasticity indicator [14,16]. Still, this does not necessarily mean the SCSVs of these 'blank' regions are not sensitive to human land use activities. More effort should be devoted towards developing a comprehensive indicator to identify the sensitivity of SCSV in response to LUCC.

## 6. Conclusions

We analyzed the characteristics of LUCC in the TGRA for 1995–2015 and its four 5-year sub-periods, mapping the SCSV in monetary units due to LUCC using the value transfer method and identifying sensitive areas of SCSV to LUCC. All outputs are available in a spatially explicit form. Three major findings were supported. (i) The expansion of construction land, water body, and reduced cropland are the most notable LUCC characteristics across the four periods. Drivers for this included the relocation of residents, construction of the dam project, and interception of Yangtze River to store water in the reservoir, and the implementation of the Grain for Green Program. (ii) The SCSV was generally stable for 1995–2015, indicating a harmonious human–environment relationship in the TGRA. The SCSV for the head part and several counties of the tail part of the TGRA were high, but low for its mid-west and regions along the Yangtze River. SCSV was concentrated in regions with elevations of 400–1600 m and slopes of $0°–10°$. The SCSV for Chongqing and its surrounding regions decreased significantly during 1995–2015. (iii) High sensitivity of SCSV in response to LUCC was found in Chongqing and its surrounding regions, where urbanization is intense and populations are dense. Fourteen counties (two-thirds of all in TGRA) had a sensitivity index that increased over the study period.

Using NDVI to spatialize the EVF can be widely used in future ES monetary mapping studies. The results of this study can help governments determine where protection and restoration is economically most important and to make better policy concerning ecological compensation and conservation.

**Author Contributions:** Conceptualization, S.L., Z.B. and G.J.; Formal analysis, Z.B.; Funding acquisition, G.J.; Methodology, S.L., Z.B. and G.J.; Resources, G.J.; Supervision, S.L. and G.J.; Visualization, S.L. and Z.B.; Writing—Original Draft, S.L. and Z.B.; Writing—Review and Editing, S.L. and G.J.

**Funding:** This research was funded by the National Key Research and Development Program of China on Global Change (No. 2017YFA0603304), the National Natural Science Foundation of China (No. 41,701,228 and 41,501,593), and Fundamental Research Funds for the Central Universities, China University of Geosciences (Wuhan), grant numbers CUGL170823 and CUG170105.

**Acknowledgments:** Many thanks to Xinliang Xu for sharing the China's Land-Use/cover datasets and NDVI datasets, which were downloaded from the Resource and Environmental Data Cloud Platform (http://www.resdc.cn/). We are grateful to three anonymous reviewers and Wangjun Li (School of Environmental Science and Engineering in Suzhou University of Science and Technology) and Qinghai Deng (College of Earth Science and Engineering in Shandong University of Science and Technology) for their valuable comments and suggestions to improve this study.

**Conflicts of Interest:** The authors declare no conflict of interest.

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
