# Peer review of "Spatially Explicit Mapping of Soil Conservation Service in Monetary Units Due to Land Use/Cover Change for the Three Gorges Reservoir Area, China"

_remotesensing, doi:10.3390/rs11040468_

Round 1

Reviewer 1 Report

Dear Editor,

Despite I did not get detailed statements on the way the authors responded to the authors' queries, I find the paper has been much improved and suggest to accept it in its present form.

Congratulations to the authors

With all the best

Author Response

Response to Reviewer 1 Comments

Point 1: Dear Editor,

Despite I did not get detailed statements on the way the authors responded to the authors' queries, I find the paper has been much improved and suggest to accept it in its present form.

Congratulations to the authors

With all the best

Response 1: Thank you for this comment.

Reviewer 2 Report

In general, this manuscript was substantially improved.

The abstract was totally reformulated. Now it is much more clear.

The Introduction was improved. However, some sentences need to be reformulated.

Fig. 1 was improved.

The upper boxes of figure 2 need to be improved. Are hard to understand.

Several doubts related to section 3.4.

Some clarifications are needed regarding the data used (section 3.5).

The results section was totally reformulated. However, several information is given with much detail.

Several minor comments could be found in the PDF file.

Author Response

Response to Reviewer 2 Comments

Point 1: In general, this manuscript was substantially improved.

Response 1:  Thank you for this comment.

Point 2: The abstract was totally reformulated. Now it is much more clear.

Response 2: Thank you for this comment.

Point 3: The Introduction was improved. However, some sentences need to be reformulated.

Response 3:  Following this comment and several minor comments in the PDF file, we reformulated some sentences in Introduction.

Point 4: Fig. 1 was improved.

Response 4: Thank you for this comment.

Point 5: The upper boxes of figure 2 need to be improved. Are hard to understand.

Response 5: Following this comment, we revised the two upper boxes of figure 2.

Point 6: Several doubts related to section 3.4.

Response 6:  Following this comment and several minor comments in the PDF file, we revised doubts related to section 3.4

Point 7: Some clarifications are needed regarding the data used (section 3.5).

Response 7: Following this comment and several minor comments in the PDF file, we revised section 3.5. The data origin of DEM was provided.

Point 8: The results section was totally reformulated. However, several information is given with much detail.

Response 8: Thank you for this comment. Based on minor comments in the PDF file, we revised our manuscript.

Point 9: Several minor comments could be found in the PDF file.

Response 9: Thank you for the line to line comments in the PDF file. Based on these comments, we revised our manuscript carefully.

This manuscript is a resubmission of an earlier submission. The following is a list of the peer review reports and author responses from that submission.

Round 1

Reviewer 1 Report

Review comments on remotesensing-412458-peer-review-v1 (1) (1)

Dear Editor,

This study has merit but I’m afraid the paper still requires thoroughly editing to reach the level of international publications and before publication is granted. The main issue is convincing of the novelty of the research and discussing it in respect of the existing research on the subject. This is missing in the present manuscript. Both the abstract and the introduction sections should convince on the novelty of the work and this can only be done by (1) acknowledging the existing literature on the subject; what has be done so far on the subject by referring to existing research studies with quantitative information, and reporting ones; referring to methods and ideas associated with other researchers; (2) discussing the existing finding and identifying research gap(s); critically discuss these existing works to summarize research advances and  gaps in knowledge of the subject (3) clearly state the research objectives, which should be in accordance with the identified gaps. Finally, the discussion section should compare the obtained results with the existing literature (the ones cited in the introduction section but not only). How do the obtained results compare to these obtained in similar regions or elsewhere ? on other objects? What are the possible explanations of the trends? What do we learn from the results?

As it stands the paper looks rather technical with no validation of the NDVI and LUCC estimations (please see for instance Chaplot and Seyler, 2001 for Landsat mapping validation using field observations), and consist of the application of existing knowledge and procedures to a new geographical area. How can the authors go further?

Another issue concerns the English grammar with few examples given below:

Introduction

Line 85: What we can concluded (conclude) from above studies is that

Mat and meth

Line 212: on this idea, we defines sensitive areas of SCSV

Line 285: And the driving factors includes (include) relocation of residents

Line 296: River is also low. In addition, it can also be seen that regions with great slope (high mean slope gradient)

Line 298: own high SCSV (are characterised by high SCSV)

Line 359: But we need to pay much 359 attention that the amplitude of the (pay attention to)

Line 437 are summarized as follows. (there should be a “:” after follows)

Finally, the abstract needs to be more informative. There are many different ways of writing an abstract and an Introduction. This depends on the academic subject involved, the journal itself and the specific topic of the article. It is important for the purpose of the research that authors can identify the patterns used in abstracts of comparable articles published in the same area, and for journals that authors might write for.

Abstract

A.        Topic sentence (s) on the subject (its importance) and research question(s): what is(are) the research gaps in this field of research?

B.        Objectives of the study

C.        Materials and methods used in the study

D.        Main results (with quantitative information, tests of significance)

E.         Conclusions:  how these results respond to the objectives; general implications of the research

Reviewer 2 Report

This manuscript analyse soil conservation service values (SCSV) spatial-temporal changes in the Three Gorges Reservoir area, China. Land cover changes and ancillary information were employed to assess SCSV evolution and its relation with the construction of the dams and subsequent socioeconomic and environmental changes. The manuscript is interesting and the presentation of the document is very good.

However, specific merits of remote sensing processing can be improved and relevant technical information was missed. NDVI images have to be processed from original remote sensing images. Resampling from 1 km to 100 m is not reasonable. All the analyses should be redone with the new NDVI dataset. In addition, several topographic parameters have been employed for the analyses but no description about a DEM was included. I include some specific comments.

Figure 1. What are the minimum and maximum altitudes of the study area? The elevation range of Figure 1 starts with a negative value. Is there any part of the study area below sea level? Please, check it.

Lines 129-132. The remote sensing information (i.e., land  cover and NDVI) employed in the manuscript was obtained from previously processed datasets.  In this sense, the specific merits of remote sensing processing can be improved. In addition, resampling NDVI products from 1 km to 100 m is highly controversial. I request the computation of NDVI values from original Landsat (or similar) datasets. After that, resampling from 25-30m to 100 m is reasonable in order to reach spatial resolution coherence for all your input information.

Line 132. You employed a digital elevation model through the manuscript (e.g., Figure 1, Table 2, or Table 3). You omitted any kind of information about it. Please, include a comprehensive description of your DEM. Data source, processing methods and spatial resolution information is mandatory.

Figure 3. Figure caption should include a reference to the original dataset of the LULC product [31].

Lines 384 and 416. Soil instead of soli.

Reviewer 3 Report

General comments

The English should be improved.

The sentence structure has to be reviewed.

The work innovation (in the scientific field) needd to be highlighted.

The dataset used are already available (Li et al.). Moreover, several details critical for the further analysis are missing in the methods section.

Most of the assumptions on which this study is based are from other works.

This study is very local.

In addition to all the problems founded, this paper does not have any content related to RS. Maybe could be submitted to Sustainability or Environments.

Specific Comments

The abstract needs to be reformulated.

The Introduction section should be carefully revised.

The legend of figure 1 (elevation) needs to be corrected.

The framework presented in Fig. 2 is very confusing.

The analysis given in section 4.2 seems to be a “free” and “subjective” analysis if the quantitative results given in table 1.

Fig. 7 is not interpretable.

Several comments are given in the pdf file.
